# Lead-OR: A multimodal platform for deep brain stimulation surgery

**Simón Oxenford[1]\*[†], Jan Roediger[1,2], Clemens Neudorfer[1,3,4], Luka Milosevic[5,6], Christopher Güttler[1], Philipp Spindler[7], Peter Vajkoczy[7], Wolf-Julian Neumann[1], Andrea Kühn[1], Andreas Horn[1,3,4]**

[1]Movement Disorders and Neuromodulation Unit, Department of Neurology, Charité — Universitätsmedizin Berlin, corporate member of Freie Universität Berlin and Humboldt Universität zu Berlin, Berlin, Germany; [2]Charité — Universitätsmedizin Berlin, Einstein Center for Neurosciences Berlin, Berlin, Germany; [3]Center for Brain Circuit Therapeutics Department of Neurology, Brigham & Women's Hospital, Harvard Medical School, Boston, United States; [4]MGH Neurosurgery & Center for Neurotechnology and Neurorecovery (CNTR) at MGH Neurology Massachusetts General Hospital, Harvard Medical School, Boston, United States; [5]Institute of Biomedical Engineering, University of Toronto, Toronto, Canada; [6]Krembil Brain Institute, University Health Network, Toronto, Canada; [7]Department of Neurosurgery, Charité — Universitätsmedizin Berlin, Berlin, Germany

**\*For correspondence:**
simon.oxenford@charite.de

**Present address:** [†]Movement Disorders and Neuromodulation Unit, Department for Neurology, Charité — Universitätsmedizin, Berlin, Germany

## Abstract

**Background:** Deep brain stimulation (DBS) electrode implant trajectories are stereotactically defined using preoperative neuroimaging. To validate the correct trajectory, microelectrode recordings (MERs) or local field potential recordings can be used to extend neuroanatomical information (defined by MRI) with neurophysiological activity patterns recorded from micro- and macroelectrodes probing the surgical target site. Currently, these two sources of information (imaging vs. electrophysiology) are analyzed separately, while means to fuse both data streams have not been introduced.

**Methods:** Here, we present a tool that integrates resources from stereotactic planning, neuroimaging, MER, and high-resolution atlas data to create a real-time visualization of the implant trajectory. We validate the tool based on a retrospective cohort of DBS patients (N = 52) offline and present single-use cases of the real-time platform.

**Results:** We establish an open-source software tool for multimodal data visualization and analysis during DBS surgery. We show a general correspondence between features derived from neuroimaging and electrophysiological recordings and present examples that demonstrate the functionality of the tool.

**Conclusions:** This novel software platform for multimodal data visualization and analysis bears translational potential to improve accuracy of DBS surgery. The toolbox is made openly available and is extendable to integrate with additional software packages.

**Funding:** Deutsche Forschungsgesellschaft (410169619, 424778381), Deutsches Zentrum für Luft- und Raumfahrt (DynaSti), National Institutes of Health (2R01 MH113929), and Foundation for OCD Research (FFOR).

## Editor's evaluation

The authors present a software tool combining and correlating the documentation of intraoperative neurophysiological findings with atlas and imaging data. They also show an exemplary validation

of their tool in a clinical series of 52 Parkinson's disease patients who underwent DBS surgery. This article will be of interest to clinicians and researchers who are involved in both the placement and controlling of the accuracy of the location of deep brain stimulation electrodes.

## Introduction

During deep brain stimulation (DBS) surgery, different sources of information are used to ensure precise placement of the electrodes within the target structure. Functional stereotactic coordinates (defined relative to anatomical atlas landmarks) are often used as a starting point (indirect targeting). Then, more importantly, preoperative MRI sequences optimized to visualize target structures are used to refine the initial plan (direct targeting). Surgical planning is usually carried out after fusing the MRI sequences with a computed tomography (CT) volume acquired with the stereotactic frame and fiducial plates already mounted to the patient's head. The fiducial plates include markers that are used to convert stereotactic coordinates (established in the planning software) to frame coordinates (applicable to mechanically adjust the stereotactic frame) in order to place electrodes to the intended target.

During the surgical procedure, microelectrode recordings (MERs), as well as test stimulations carried out using macroelectrodes, are often used as an additional confirmation step of placement in the intended target site. While the necessity of the former step has been debated (*Aviles-Olmos et al., 2014*) and the procedure may lead to slightly increased rates of complications (*Zrinzo et al., 2012*), the experience of our own high-volume center is that roughly every fifth patient's surgical plan will be slightly altered based on electrophysiological signals, with similar experiences reported by others (*Lozano et al., 2018*). Of specific relevance is the role of brain shift occurring due to air entering the skull during surgery: even with optimal imaging and meticulous surgical planning before-hand, brain shift may lead to nonlinear displacement of the brain relative to the skull and stereotactic frame (*Halpern et al., 2008*), which can only be monitored intraoperatively (e.g., using the electrophysiological data recorded with microelectrode probes). While most centers analyze MERs by visual and auditory inspection from expert neurosurgeons or neurologists, the first FDA and CE-approved machine-learning algorithms that facilitate this monitoring step have recently been introduced, for instance, in the form of the HaGuide system created by the company Alpha Omega Engineering (Nazareth, Israel; *Thompson et al., 2018*).

Still, understanding and communicating the complex neuroanatomical and neurophysiological relationships within the clinical team during the procedure may remain a challenge even for experts. To account for this, *Krüger, 2020* introduced the concept of navigated DBS surgery, showing that a visual feedback of the microelectrode position can be helpful to mentally envision the ongoing 3D scene.

In parallel, reconstructions of DBS target regions based on elaborate MRI sequences have become increasingly precise (*Horn, 2019*; *Krauss et al., 2021*). Specialized MRI sequences have been introduced to maximize visibility and boundary definitions of pallidal, thalamic (*Tourdias et al., 2014*; *Sudhyadhom et al., 2009*; *Vassal et al., 2012*), and subthalamic (*Santin et al., 2017*; *Wang and Liu, 2015*) targets. But even when relying on a set of standard sequences (e.g., T1 and T2), modern reconstruction pipelines have the capability to reconstruct the subthalamic nucleus (STN) and internal segment of the globus pallidus (GPi) with a precision that rivals manual expert segmentations (*Ewert et al., 2019*). Over recent years, these methods have made it possible to transform the 2D representations of stereotactic imaging slices into 3D models that are not only graphically appealing but indeed realistic and meaningful (*Horn and Kühn, 2015*). As a by-product, these tools have made it possible to accurately register atlas data into the patient-specific model. With *atlas data*, here, we generally refer to an array of ultra-high-resolution imaging resources that could be based on histology (*Ilinsky et al., 2018*; *Ewert et al., 2018*; *Amunts et al., 2013*), postmortem MRI (*Edlow et al., 2019*), or even expert anatomical knowledge aggregated in three-dimensional fashion (*Petersen et al., 2019*). Similarly, atlas data could represent optimal stimulation sites defined on a group level, for instance, in the form of probabilistic sweet spot targets (*Dembek et al., 2019*; *Boutet et al., 2021*; *Elias et al., 2021*; *Horn et al., 2017*) or tractography-defined DBS target atlases (*Li et al., 2020*; *Treu et al., 2020*; *Al-Fatly et al., 2019*).

**eLife digest** Deep brain stimulation is an established therapy for patients with Parkinson's disease and an emerging option for other neurological conditions. Electrodes are implanted deep in the brain to stimulate precise brain regions and control abnormal brain activity in those areas. The most common target for Parkinson's disease, for instance, is a structure called the subthalamic nucleus, which sits at the base of the brain, just above the brain stem.

To ensure electrodes are placed correctly, surgeons use various sources of information to characterize the patient's brain anatomy and decide on an implant site. These data include brain scans taken before surgery and recordings of brain activity taken during surgery to confirm the intended implant site. Sometimes, the brain activity signals from this last confirmation step may slightly alter surgical plans. It represents one of many challenges for clinical teams: to analyse, assimilate, and communicate data as it is collected during the procedure.

Oxenford et al. developed a software pipeline to aggregate the data surgeons use to implant electrodes. The open-source platform, dubbed Lead-OR, visualises imaging data and brain activity recordings (termed electrophysiology data) in real time. The current set-up integrates with commercial tools and existing software for surgical planning.

Oxenford et al. tested Lead-OR on data gathered retrospectively from 32 patients with Parkinson's who had electrodes implanted in their subthalamic nucleus. The platform showed good agreement between imaging and electrophysiology data, although there were some unavoidable discrepancies, arising from limitations in the imaging pipeline and from the surgical procedure. Lead-OR was also able to correct for brain shift, which is where the brain moves ever so slightly in the skull.

With further validation, this proof-of-concept software could serve as a useful decision-making tool for surgical teams implanting electrodes for deep brain stimulation. In time, if implemented, its use could improve the accuracy of electrode placement, translating into better surgical outcomes for patients. It also has the potential to integrate forthcoming ultra-high-resolution data from current brain mapping projects, and other commercial surgical planning tools.

Here, we present an integrative approach to combine information derived from neuroimaging and neurophysiology in a joint visualization platform. First, we build on recent validations of subcortical normalization routines to introduce a method to refine 3D models of subcortical targets on a single patient level. Second, we port our methodology for postoperative electrode localization established within Lead-DBS software (https://www.lead-dbs.org; *Horn and Kühn, 2015*) to the pre- and intraoperative realm, that is, the one of stereotactic planning, MERs, and intraoperative testing. To achieve this, we present and validate a novel unified software framework termed Lead-OR that incorporates the following resources into a live visualization scene: (1) patient-specific imaging, (2) stereotactic planning information, (3) real-time microelectrode localization, (4) MER feature extraction, and (5) high-resolution atlas imaging data. The capability of the system to integrate electrophysiological information with imaging data is explored in-depth. Beyond this feature, the tool also includes the possibility to visualize test stimulations and real-time fiber tractography. The software framework is made available as an open-source package (https://github.com/netstim/SlicerNetstim) and currently supports integration with the Brainlab Elements (Brainlab AG, Munich, Germany) planning software and a direct interface to the NeuroOmega system (Alpha Omega Engineering). Further integrations with other systems are planned in the future.

## Methods
### Ethics statement

Lead-OR is intended for purely academic research use and does not have any form of government body regulatory approval. As such, any use of Lead-OR is strictly limited to Institutional Review Board (IRB)-approved research studies at individual academic institutions, while legal frameworks and practices may differ from country to country. The collection and analysis of all patient data used for this article were approved by the local ethics committee of Charité – Universitätsmedizin Berlin (master vote EA2/145/21). All data were analyzed retrospectively and obtained in deidentified form

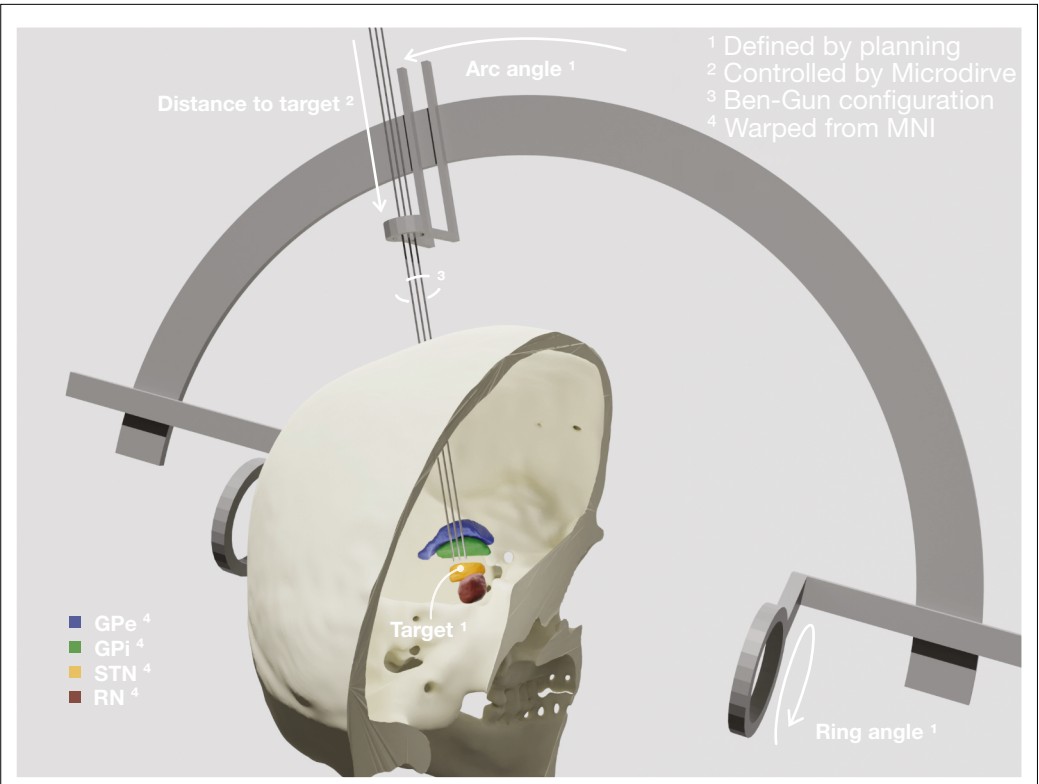

**Figure 1.** Patient-specific visualization generated by aggregating different sources of data. The stereotactic planning procedure defines the surgical target coordinate, as well as ring and arc angles, which together describe the central trajectory. The Ben-Gun configuration presented in the figure shows additional posterolateral and anteromedial trajectories, 2 mm apart from the central one. Up to five trajectories are currently supported by the software. In our current setup, the distance to the target is controlled by the NeuroOmega system, accessed with its Software Development Kit (SDK) – but can alternatively be set manually within the tool itself. Relevant subcortical nuclei have been warped to patient space via a manually refined normalization. GPe: external segment of the globus pallidus; GPi: internal segment of the globus pallidus; STN: subthalamic nucleus; RN: red nucleus.

from Medical Records of Charité. Hence, following local guidelines in Berlin/Brandenburg as well as NIH guidelines for human subjects research, no explicit patient consent to analyze and publish was obtained/necessary.

## Implementation environment

The tools used in this study are implemented in the form of a 3D Slicer (Slicer) (*Fedorov et al., 2012*; *Kapur et al., 2016*) extension (https://github.com/netstim/SlicerNetstim). The main module of the SlicerNetstim extension is Lead-OR, which assembles the different sources of information, as outlined in the following sections.

## Coordinate systems

The first step in aggregating data from different sources is to co-register their spatial relationship and coordinate systems (*Figure 1*). Lead-OR is based on Slicer's world-coordinate system (RAS). We use a linear transform to match the Head-Ring center and positive axes to the origin of this world-coordinate system. The planned central trajectory is then defined based on target coordinates, mounting type, and ring and arc angles. The other trajectories are defined relative to the central one, following the configuration of the Ben-Gun microarray. As mentioned, currently, support for the NeuroOmega setup has been implemented, which uses a Ben-Gun array first introduced by the team of Alim-Louis Benabid (*Benazzouz et al., 2002*).

These trajectories describe a line in space through which the macro, micro, and definitive DBS electrodes are inserted. The last parameter to fully define their position varies throughout surgery,

namely, the distance to the planned target. This parameter is set by the Microdrive, which allows to move the electrodes along the trajectories while recording from the tip of the microelectrodes. In our current setup, this value is queried via the NeuroOmega Software Development Kit (SDK) and alternatively can be manually controlled within the software itself. Interfacing to similar systems as the NeuroOmega device will be possible given the open-source nature of our tool and creating such interfaces with other systems is in our interest for the future.

To co-register the patient's images and frame reference, the tool uses a set of fiducial points defined in both coordinate systems (image and frame) that we extract from the surgical planning coordinates. Specifically, the anterior commissure (AC) and posterior commissure (PC), as well as a midsagittal point (MS), are used to create the transform (implemented using the fiducial registration module available within Slicer). Currently, an interface with the Brainlab Elements (Brainlab AG) stereotactic planning software is implemented (via PDF export in Elements and automated import in Lead-OR). Again, support for alternative planning tools is planned for the future.

Finally, we incorporate high-resolution atlas resources into the patient-specific visualization scene. For the present examples within the article, we used nuclei from the DISTAL (*Ewert et al., 2018*) and MNI PD 25 histology atlases (*Xiao et al., 2017*) that were defined in MNI space (ICBM 2009b Nonlinear Asymmetric, *Fonov et al., 2009*). Similarly, we imported histological sections from the BigBrain atlas (*Amunts et al., 2013*) and fiber tract definitions provided by the holographic basal ganglia pathway atlas (*Petersen et al., 2019*). In the same fashion, virtually any type of atlas data could be imported to the patient scene, but it is crucial that this registration is of utmost precision. To account for this, we built on the long-standing methods development within Lead-DBS (*Horn and Kühn, 2015*; *Horn et al., 2019*; *Ewert et al., 2019*; *Vogel et al., 2020*; *Edlow et al., 2019*) but drastically extended the procedure with a novel manual refinement method, termed WarpDrive. Namely, an initial deformation field was calculated via a multispectral four-stage normalization step using the symmetric normalization (SyN) transformation model implemented within Advanced Normalization Tools (ANTs; http://stnava.github.io/ANTs/; *Avants et al., 2008*). This was implemented using the 'effective: low variance + subcortical refinement' preset defined in Lead-DBS, which has been optimized for normalization of subcortical structures (*Horn et al., 2019*) and has shown to yield accurate

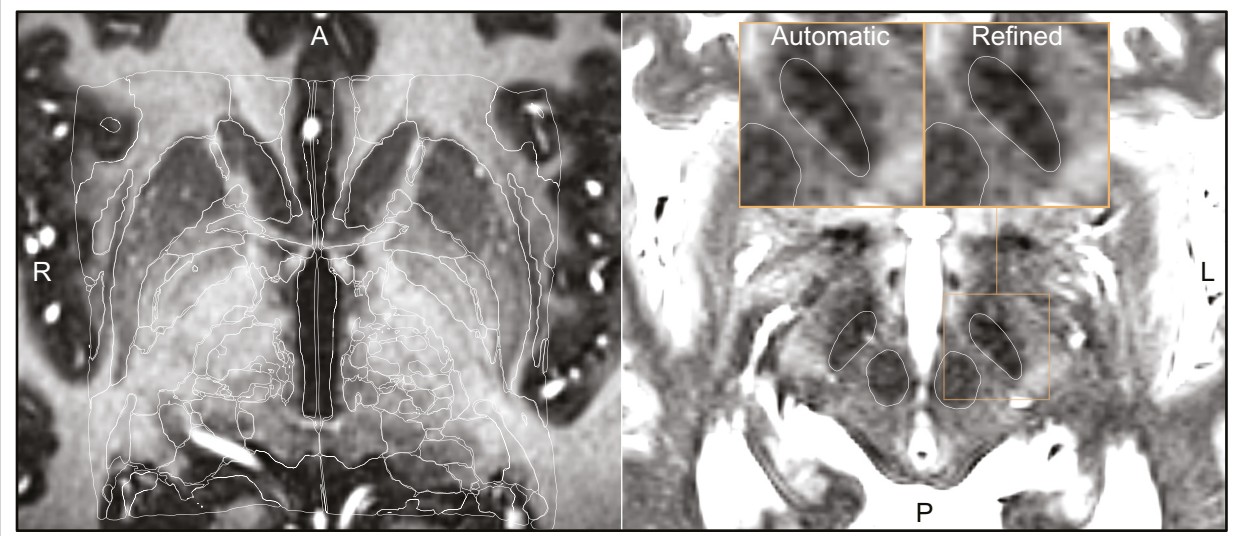

**Figure 2.** Output of a Lead-DBS/Advanced Normalization Tools (ANTs)-based automated normalization with and without subsequent manual refinement. Two MRI modalities are shown anterior commissure-posterior commissure (AC-PC) aligned: T1-MPRAGE (left) and T2-TSE (right). Both MRI modalities (together with FGATIR, not shown here) were used as an input to the normalization step implemented in Lead-DBS, which allows multispectral registration using ANTs. The white outline shows atlases: MNI PD 25 histology (*Xiao et al., 2017*) (left) and DISTAL (*Ewert et al., 2018*) (right), both included within Lead-DBS.

The online version of this article includes the following video for figure 2:

**Figure 2—video 1.** General overview of the visualizations and tools made available through the WarpDrive module implementation in Slicer.
https://elifesciences.org/articles/72929/figures#fig2video1

segmentations of subcortical nuclei that rival the ones carried out manually by experts (*Ewert et al., 2019*; *Vogel et al., 2020*). The deformation fields derived from this automated step are then further manually refined using WarpDrive, which is described in the next section.

## Normalization refinement

While normalization algorithms have become increasingly accurate (*Vogel et al., 2020*; *Ewert et al., 2019*), their precision is not always perfect in single subjects and shows varying accuracy throughout the brain. Indeed, accurate automated registration of the basal ganglia nuclei presents a challenge to intensity-based registration methods given their low contrast between regions (*Ewert et al., 2019*).

Using WarpDrive, an experienced user can recognize such mismatches included in the automated normalization and manually refine the displacement field using point-to-point and line-to-line fiducials as well as a smudge tool. Manually entered fiducials are fed into the Plastimatch software (*Sharp et al., 2010*) (accessed as a command line module from within Slicer). Details about the WarpDrive tool will be reported and evaluated elsewhere. *Figure 2* shows an example of a manually refined normalization, and *Figure 2—video 1* shows a demo application of the tool to refine atlas-to-patient fits in a surgical case.

## Real-time implementation

Lead-OR has the potential to be used in real-time during surgery. As mentioned above, one aspect of this is the continuous/live updating of the microelectrode distance to the surgical target while keeping the scene (i.e., multiple 2D and 3D views) synchronized. The interface to the NeuroOmega device

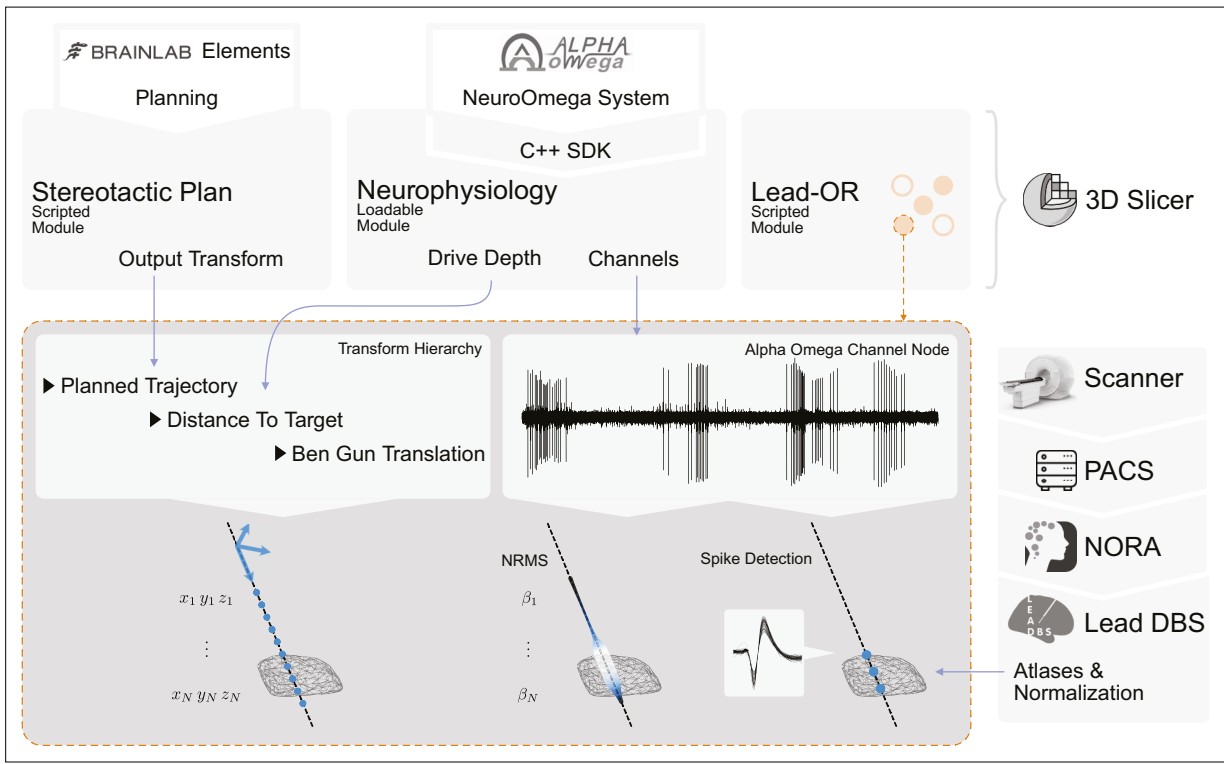

**Figure 3.** Overview of the SlicerNetstim extension. The current setup shows interfaces with specific commercial products. Similar interfaces to competing tools are planned and will be included in the future. A PDF plan exported from Brainlab Elements is used as an input to the Stereotactic Plan module to store the planned trajectory as a Slicer Transform. The NeuroOmega system is connected via its Software Development Kit (SDK) through the Neurophysiology module, providing continuous updates about the drive depth and electrophysiological channel input. Finally, in the Lead-OR module, the Ben-Gun configuration is defined by selecting the used trajectories and assigning them to input channels from the NeuroOmega device. Using a transform hierarchy, the spatial position of the microelectrode is defined: the Ben-Gun translation is transformed by the distance to the target, this one itself being transformed by the planned trajectory. By doing so, the features extracted from the respective microelectrode recordings (MERs) can be mapped to their spatial location. At our center, an automatic pipeline for preprocessing data retrieved from a picture archiving and communication system (PACS) system is setup using the NORA medical imaging platform (https://www.nora-imaging.com/) to automatically run the core Lead-DBS pipeline once images arrive in the hospital's PACS system. This part (right-hand side) is not discussed in detail since it is largely specific to our center.

provides live data about how distant the microelectrodes are to the target and also streams out electrophysiological recordings made in a real-time manner. Finally, test stimulations can be visualized, including a function for live-tractography visualization estimating 'activated' or 'modulated' tracts.

To make this possible, we included the NeuroOmega C++ SDK as part of a Slicer loadable module. This module sets up the connection to the NeuroOmega device and queries the distance to the target in specified time intervals. It also displays the available channels from which recorded electrophysiological data can be streamed, stored, processed, and visualized.

Through the Lead-OR module, the microelectrode Ben-Gun configuration is defined and the NeuroOmega channels are linked to the selected trajectories. Together with the aforementioned image-to-frame transform, as well as the distance to the target, this allows to define the anatomical location of the electrophysiological signal in real time. The features extracted from recordings are projected into the patient-specific space and represented in the 2D/3D visualization (*Figure 3*). The computation of features is continuously executed for each position of the microdrive, updating the recording stream at each time point. This process takes a few seconds (depending on available hardware), and the visualization is then updated.

(Re-)developing a signal processing pipeline for electrophysiological data was not the focus of this study since a multitude of tools exist, which could be integrated into Lead-OR in the future. However, to demonstrate live processing and visualization of electrophysiological features, for now we included two minimal processing pipelines for MER. (Currently, no pipeline for local field potential recordings is included, but this could be similarly extended given the open-source nature of the tool.).

The first is the signal's normalized root mean square (NRMS) value, which is computed as described in *Zaidel et al., 2009*. For each step (Microdrive position), a stable part of the recorded data is extracted to compute the RMS on (see Zaidel et al., supplementary material). To obtain a normalized measure, the values along the trajectory are divided by the median of the first five stable steps. To visualize results in space, Lead-OR projects a tube along the trajectory with varying radius and color – both redundantly representing NRMS magnitude. Potentially, in the future, radius and color could be assigned to represent different features that could graphically combine information derived from MER and local field potential signals.

The second processing pipeline is based on spike analysis. This is implemented by running the WaveClus (*Chaure et al., 2018*) automatic pipeline with negative threshold on the recorded files once the drive moves to the next position. Clusters with less than 100 spikes or in which 10% of the inter spike intervals (ISIs) are below 3 ms or in which signal-to-noise ratio (SNR) is less than 1.5 are discarded. SNR is computed as described in *Joshua et al., 2007* using the residual method. We assume the recordings capture singe-unit activity (SUA) instead of multiunit activity (MUA), and thus each recording can represent none or one cluster of spikes. One of the reasons behind this assumption is, for example, that changes in amplitude recording from the same unit might be misclassified as different clusters. Spike clusters are represented as fiducials placed in the position they were detected. *Figure 3* summarizes the described live-processing setup.

## Stimulation module

Intraoperative assessment of stimulation-induced therapeutic as well as side effects can yield important information about electrode placement. For example, electrode placement close to the internal capsule may lead to tonic muscle contractions at low stimulation amplitudes. Often, these thresholds are intraoperatively identified by stimulating at increasing steps until muscle contractions and/or electromyography (EMG) activity are observed. Since Lead-OR already visualizes the patient-specific location of the stimulation sites, volumes of tissue activated (VTA) could be used as seeds for tractography. Fiber analysis was carried out by accessing the logic of the SlicerDMRI module (*Norton et al., 2017*).

Obtaining preoperative diffusion MRI data is not part of clinical practice at all DBS centers. In cases where patient-specific dMRI data is not available, an alternative is to use normative fibers that are defined in template space and warped into patient space (similar to other types of atlas data). This process can at times even come with advantages, for example, the absence of false-positive fibers when using manually curated normative datasets (*Petersen et al., 2019*; for a more thorough discussion, see *Horn and Fox, 2020*). For the purpose of this article, we will refer to the term tractography as the process to filter and visualize tracts derived from such normative datasets or whole-brain

tractography connectomes (*Reisert et al., 2011*) that intersect with a region of interest (ROI) (in our case, the VTA). The exact same process is possible using patient-specific streamlines but is not demonstrated here.

To estimate the VTA, we use the simplified method proposed by *Dembek et al., 2017*, which defines the radius of a sphere based on stimulation amplitude and pulse width. Varying values from 0.5 to 1.0 were set for the constant $k_2$ in their formula (see Dembek et al., supplementary material, for the explanation of the parameter). In our present example, a value of $\sim 0.8$ seemed to yield results that matched the recorded EMG data. Currently, this part of this article should be seen as exploratory as an example of feasibility. Data to validate the approach on a larger number of patients beyond the present case was lacking. Further studies are needed to titrate the $k_2$ value on a group level and validate the stimulation module of Lead-OR in general.

## Patient cohort and surgical procedure

Up to this point, we described the live setup of Lead-OR. We aimed to evaluate the accuracy of this setup by comparing imaging- and electrophysiology-derived markers on a group level. To do so, we retrospectively gathered data from patients who underwent DBS and processed it in a similar fashion as the real-time application. 52 patients (12 females; mean age $= 61 \pm 9$) were retrieved from cases undergoing STN-DBS surgery at Charité – Universitätsmedizin Berlin between 07/2017 and 10/2021. Inclusion was based on having homogeneous data acquisitions consistent with current surgical procedure (i.e., Brainlab planning exports together with corresponding imaging acquisitions and complete microelectrode recording information). *Supplementary file 1* summarizes the inclusion process in the form of a flow chart.

Patients underwent bilateral DBS surgery targeting the STN. Surgery was either performed awake or under general anesthesia. In case of the latter, the depth of narcosis was reduced before MERs to reduce potential effects of anesthetic drugs.

The NeuroOmega System (Alpha Omega Engineering) was used with 2–5 microelectrodes in orthogonal (0°) or rotated (45°) Ben-Gun configuration to acquire MERs. Recordings were carried out from 10 mm above to 4 mm below the target with step sizes between 0.2 mm and 0.5 mm (with some exceptions common to clinical practice). Then, microelectrodes were removed and test stimulations were applied at multiple heights above the target via macroelectrodes on central and alternate trajectories. Neuroprobe Sonus non-shielded microelectrodes (Alpha Omega Engineering) were used as micro-/macroelectrodes. Stimulations were done at increasing amplitude steps of 0.5 mA until identifying permanent side effects. Additionally, therapeutic stimulation effects were evaluated when the surgery was performed in the awake state. Patients who underwent general anesthesia received additional EMG using needle electrodes to evaluate motor unit activity of eight muscles as indicator for the activation of corticobulbar and corticospinal tracts. Finally, based on imaging, electrophysiological, and clinical findings, the surgical team decided upon the optimal depth and trajectory for permanent electrode implantation.

Of the 52 patients included in this study, 4 were discarded based on poor imaging quality and 16 based on poor electrophysiology signals (both determined by visual inspection). Additionally, taking the same considerations, four left and four right hemispheres were also discarded based on a low quality of electrophysiology data. MERs were saved as segments for each distance to the target value. Segments were discarded if they were contaminated by artifacts or when their recording length was less than 4 s. With these considerations, we analyzed a final cohort of 32 patients (56 hemispheres) with a total of 236 trajectories.

## Imaging and electrophysiology processing

Pre- and postoperative imaging data were co-registered and normalized using Lead-DBS (*Horn et al., 2019*) followed by visual inspection and, if necessary, refinement using WarpDrive. The definitions of the central trajectories were extracted from stereotactic planning reports and the Ben-Gun configuration from recordings files. We computed the NRMS of all trajectories and resampled them on a linear space with 0.1 mm distance to target resolution. Spike clusters were computed as described above. As mentioned earlier, if more than one cluster was detected in a segment and satisfied the stated conditions, this was still considered an SUA (and represented as one cluster in further analysis).

Using the Lead-DBS pipeline, we carried out brain shift correction using postoperative CT data (*Horn et al., 2019*). This allowed us to quantify the degree of brain shift occurring after surgery based on imaging-derived metrics. For each trajectory, each recording position was displaced using the brain shift correction transform. We took the median of displacements as the amount of brain shift for each trajectory. We will refer to this measure as the imaging-based brain shift estimate in this article (note that it is derived from pre- and postoperative imaging data).

The most recently available clinical stimulation settings were retrieved from all 32 patients (visits ranging from 3 to 44 months after surgery). We reconstructed DBS electrodes based on the standard Lead-DBS pipeline and denoted the coordinate of the active contact (in case of multiple active contacts, their locations were averaged). For a qualitative analysis, we projected this coordinate to the nearest point along the closest trajectory for each electrode.

Each recording segment had its own patient-specific distance to target measure. In order to carry out group analyses, we defined a normalized distance to target. With the nonlinear deformation displacement fields, a link between the location of the trajectory and the ICBM 2009b NLIN ASYM (*Fonov et al., 2009*) ('MNI') space was established. We then took a reference point in each trajectory computed as the nearest point to the STN target coordinates in MNI space from *Caire et al., 2013*. The normalized distance to the target was defined by aligning the references of all trajectories. The alignment was done by displacing each trajectory by its reference position minus the average displacement from all trajectories (*Figure 5—source code 1*). Furthermore, by using the warp to MNI space we were able to compute the trajectory's distance to the STN and the STN entry and exit sites (henceforth referred to as imaging-defined STN boundaries). To do this, we used the STN as defined by the DISTAL atlas. The main hypothesis from the group analysis was that electrophysiological recordings acquired from within the imaging-defined STN would show higher activity than the ones recorded outside of the STN.

All spike clusters were mapped to the left hemisphere of the MNI space (right hemisphere coordinates were nonlinearly flipped). Then, we created an image of 0.22 mm isotropic resolution where each voxel represented the number of clusters detected divided by the number of segment recordings within 1 mm of the voxel's center. This resulted in a cluster density volume in MNI space (*Figure 5—source code 1*; *Figure 5—source data 2*).

Additionally, NRMS values and STN distances for each trajectory were transformed with the inverse tangent function resulting in similar distributions of the two. Subsequent cross-correlation of these two signals along each trajectory resulted in a maximum cross-correlation value and the lag (displacement) at which this maximum occurred (*Figure 5—source code 2*).

In the next step, we sorted the trajectories according to their maximum cross-correlation and split the data in half, retaining the trajectories that were in close proximity to the STN and showed electrophysiological activity. We then sorted the top half according to the lag at which the maximum cross-correlation occurred (*Figure 5—source code 2*). We will refer to this lag as the electrophysiology-based brain shift estimate (note that it is derived from preoperative imaging and intraoperative electrophysiology). Hence, in contrast to the imaging-derived brain shift estimate (which required postoperative imaging), this one could be computed during surgery. The electrophysiology-based brain shift measures were compared to the imaging-based brain shift estimates in two ways: first, we contrasted imaging-based brain shift estimates between the low versus high-lag groups using Wilcoxon's signed-rank test. The high-lag group was defined by taking trajectories with lag values above 1 standard deviation of the lags. The low-lag group is composed of the same number of trajectories taken from the data sorted according to the lag. This would analyze whether cases with high electrophysiology-derived estimates indeed had more brain shift based on the imaging-based estimate. Second, we correlated values from the high-lag trajectories (where significant brain shift was estimated based on electrophysiology) with the imaging-derived estimate of brain shift. This would analyze whether the degree of brain shift would correlate between electrophysiology- and imaging-derived estimates.

## Results

The main result of this work consists of an integrated software framework that links electrophysiological with imaging-derived data within the same patient-specific coordinate space during surgery. *Figure 4* shows the software output for a single-case example including different forms of visualization and an exemplary match between DBS imaging and electrophysiology. Furthermore,

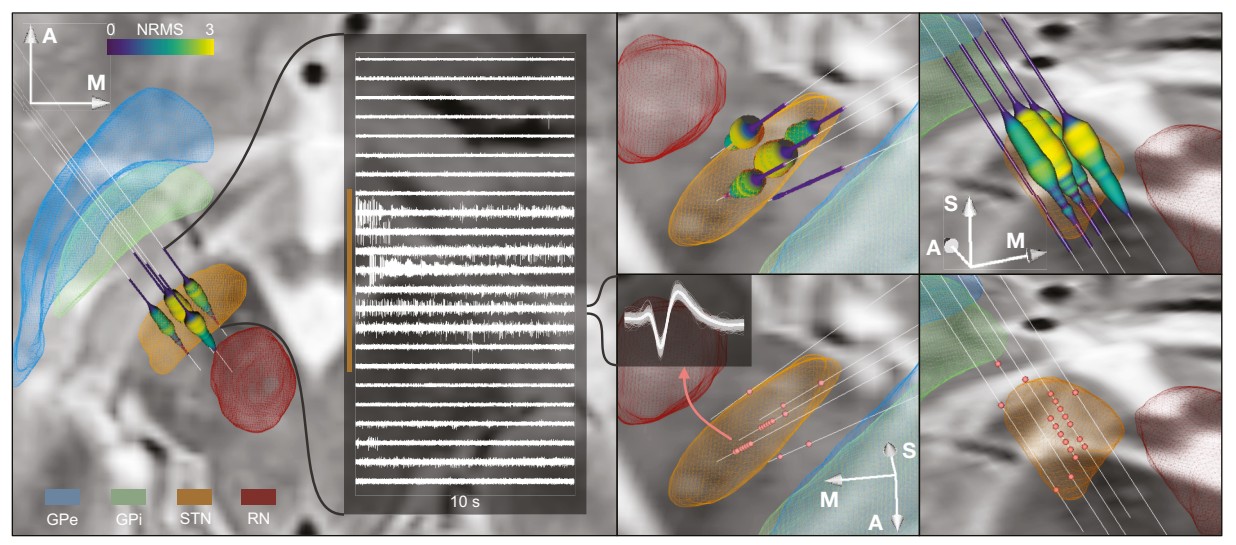

**Figure 4.** Example case showing trajectories, microelectrode recording (MER) features, and DISTAL atlas volumes mapped to patient space. 10 s recording snippets from one trajectory are displayed. Normalized root mean square (NRMS) activity is represented by a tube with varying diameter and color matching the value. Spike clusters are represented by red point fiducials. GPe: external segment of the globus pallidus; GPi: internal segment of the globus pallidus; STN: subthalamic nucleus; RN: red nucleus.

The online version of this article includes the following video, source data, and figure supplement(s) for figure 4:

**Figure supplement 1.** Additional ventral intermediate nucleus (VIM) and internal segment of the globus pallidus (GPi) cases.

**Figure supplement 2.** Additional example cases of subthalamic nucleus-deep brain stimulation (STN-DBS) Lead-OR visualizations.

**Figure 4—video 1.** General overview of the Lead-OR real-time application.

https://elifesciences.org/articles/72929/figures#fig4video1

**Figure 4—video 2.** Video showing the program user interface and its use.

https://elifesciences.org/articles/72929/figures#fig4video2

**Source data 1.** Slicer scene containing the spatial data shown in the *Figure 4*.

*Figure 4—figure supplement 1* shows the application of the tool in a ventral intermediate nucleus (VIM) and GPi example. Finally, for illustrative purposes, we included additionally three STN cases with unusual anatomical configurations in *Figure 4—figure supplement 2*. *Figure 4—video 1* shows the live application of the tool in action, and *Figure 4—video 2* shows the user interface and how the platform is setup.

*Figure 5* shows the 236 trajectories retrospectively gathered from 32 patients, arranged from left to right based on their distance to the STN and vertically aligned with the normalized distance to target. Electrophysiology traces were plotted with STN entry and exit markers derived from imaging. Comparing the NRMS from the bottom 20% (outside of the STN) to the top 20% revealed an anatomical region with significant differences (p<0.01) within the imaging-defined STN boundaries (defined as the median of the top 20% STN boundaries). In other words, the recorded activity from inside this part of the STN was significantly higher than the ones recorded outside of it. Data were compared using nonparametric Wilcoxon's signed-rank test and multiple comparisons were corrected using false discovery rate (FDR) (*Benjamini et al., 2006*).

The cluster density volume in MNI space also showed a general agreement with the imaging-derived STN: when thresholding the volume based on increasing density values, the overlap with the STN region was higher (95% overlap at a 50% cluster density threshold; *Figure 5*).

With respect to the brain shift analysis, the low-lag and high-lag groups showed a significantly different brain shift distribution (Wilcoxon's signed-rank test p=0.0076). Also, correlating the high-lag values (electrophysiology-derived brain shift estimate) with their imaging-derived brain shift estimates showed a significant association ($R = 0.40$, p=0.016; *Figure 5—figure supplement 1*). *Figure 5—figure supplement 2* shows an example case illustrating how the imaging-based brain shift-corrected Lead-OR scene presents better correspondence between imaging and MER.

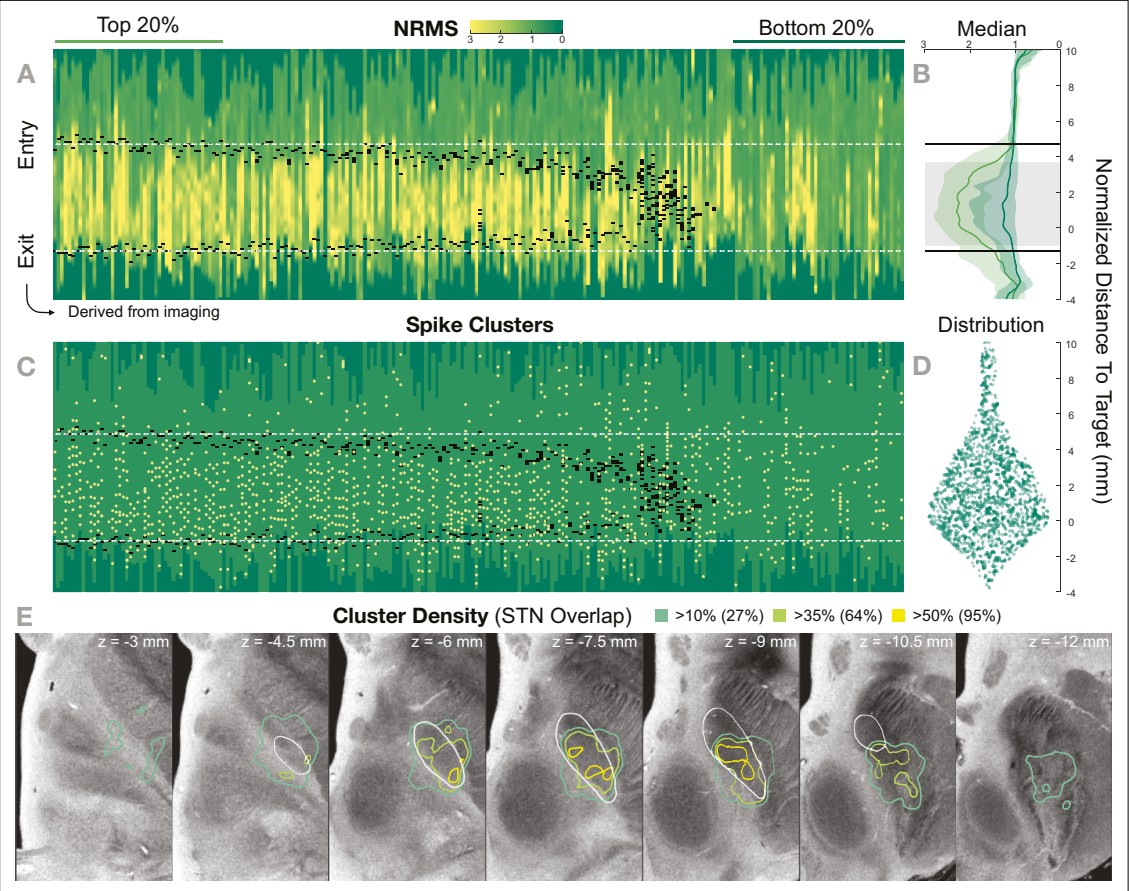

**Figure 5.** Retrospective group analysis investigating agreement between imaging- and electrophysiology-defined subthalamic nucleus (STN). In (**A**) and (**C**), each trajectory is presented as a column, showing normalized root mean square activity (NRMS) and spike clusters, respectively, with the normalized distance to target denoted on vertical axes. Trajectories are sorted from left to right according to their distance to the STN as defined in the DISTAL atlas (*Ewert et al., 2018*). Dark green values (indicating NRMS of zero) represent no recordings at these sites. Black dashes represent STN entry and exit, and the dashed white line the median entry and exit for the top 20%. (**B**) shows comparisons between bottom and top trajectories, with the gray area representing a significant band (nonparametric Wilcoxon's signed-rank test p<0.01 with false discovery rate [FDR] correction), which resides within the STN. The plots show median, 0.25 and 0.75 quantiles. (**D**) shows the overall distribution of spike clusters. (**E**) shows isosurfaces of a volume where each voxel contains the number of clusters detected divided by the number of recordings carried out within 1 mm distance to the location (cluster density). The legend shows the percentage of the volume overlap with the STN at different thresholds. The 7 T MRI ex vivo human brain template (*Edlow et al., 2019*) is shown as the background image with DISTAL STN outline. Decreasing values on the z-axis traverse from superior to inferior.

The online version of this article includes the following source data, source code, and figure supplement(s) for figure 5:

**Source code 1.** Uses *Figure 5—source data 1* to generate *Figure 5—source data 2* and panels A–D.

**Source code 2.** Uses *Figure 5—source data 1* to generate *Figure 5—figure supplement 1*.

**Source data 1.** Trajectories data including normalized root mean square (NRMS) traces, subthalamic nucleus (STN) entry-exit positions, spike clusters, brain shift, and distance to STN values.

**Source data 2.** Cluster density volume shown in Figure 5E.

**Figure supplement 1.** Brain shift study.

**Figure supplement 2.** Example case in which the imaging-derived brain shift transform was applied to the Lead-OR scene, post-hoc.

**Figure supplement 3.** Active contact coordinates overlayed with *Figure 5A and E*.

In *Figure 5—figure supplement 3*, we show clinical active contact coordinates with respect to the results of the group analysis as shown in *Figure 5*. Most of the coordinates rely inside the STN and coincide with high-activity regions as depicted by the microelectrode recordings.

*Figure 6* shows an example case using the test stimulation setup with live volume activation tractography and corresponding EMG activity invasively recorded during surgical routine using a needle

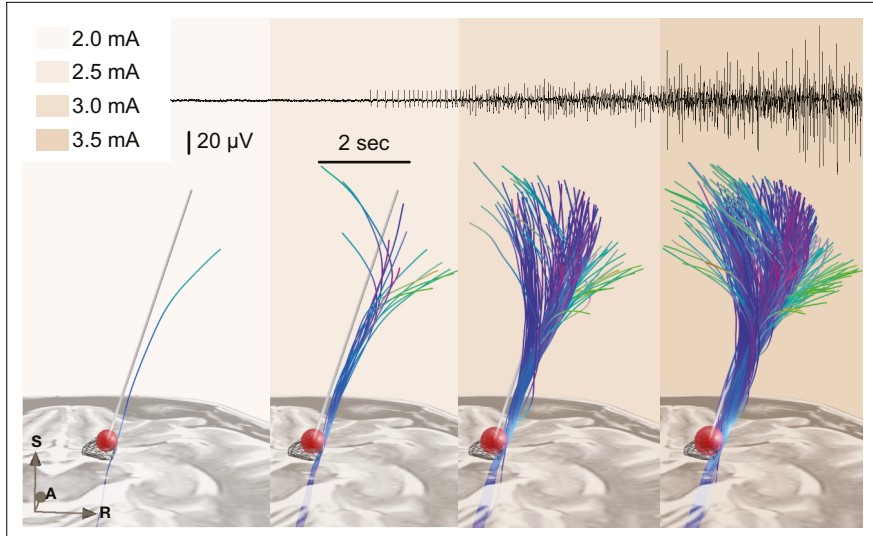

**Figure 6.** Example of test stimulation setup (also see *Figure 4—video 1* for the demonstration of a real-time application). A simplified stimulation volume is modeled based on the applied test stimulation parameters following the approach of *Dembek et al., 2017*. From a set of predefined fiber tracts representing the internal capsule (without hyperdirect components; *Petersen et al., 2019*) that were registered to patient space, fibers passing through the volume were visualized in real time. Alternatively, tractograms obtained based on diffusion MRI of the individual patient data or normative connectomes could be used. The top panel shows needle electromyography (EMG) activity that was recorded within clinical routine from the brachioradialis muscle during stimulation in the same patient. Colors represent stimulation amplitude. After a preliminary exploratory analysis of the $k_2$ parameter from Dembek's formula, a value of 0.8 was used for the shown example.

The online version of this article includes the following source data for figure 6:

**Source data 1.** Slicer scene containing the spatial data shown in the *Figure 6*.

electrode inserted into the brachioradialis muscle. We also refer to *Figure 4—video 1* for a demonstration of the real-time application of this module.

## Discussion

Multiple take-home points can be drawn from this study. First, we established a software pipeline to integrate imaging and electrophysiology results within an interactive real-time application during DBS surgery. The setup interacts with commercial tools for surgical planning and MERs and has the capability to visualize and analyze data in various forms. In the presented group study, the data acquisition conditions were not controlled for, given their retrospective nature. However, the platform can generalize to alternate settings. For example, the number of trajectories used can be set from 1 to 5, without compromising its execution. With respect to hardware settings, while currently a fixed set of interfaces to commercial tools is available, the open-source nature of the software will allow integration of links to other devices. Furthermore, although we present the tool and analysis made with STN cases, it could also be applied to other DBS targets. As illustrative examples, we refer to *Figure 4—figure supplement 1* to see Lead-OR visualizations for a VIM and GPi case. Second, atlas data from ultra-high-resolution resources may be integrated into the tool. For instance, whole-brain histological atlases, such as the BigBrain dataset (*Amunts et al., 2013*) or stereotactic 3D atlases, such as the DISTAL (*Ewert et al., 2018*) or Human Thalamus Atlas (*Ilinsky et al., 2018*) could be integrated. In a way, these atlases would fill the gap of commonly used histological reference atlases available in book format, such as the Schaltenbrandt-Wahren (*Schaltenbrand et al., 1977*) or Talairach atlases (*Rey et al., 1988*). While these book resources have been and still are invaluable to the field, they lack the possibility to be deformed into native patient space and to be digitally represented in direct synopsis with patient imaging and electrophysiology. Instead, whole-brain resources will grow in number, resolution, and quality in the foreseeable future (*Horn, 2019*; *Krauss et al., 2021*; *Sui et al., 2020*; *Vedam-Mai et al., 2021*). Similarly to anatomical atlas resources, optimal target definitions

('sweet spots') or even connectomic/tract-based target definitions could one day be integrated to guide DBS surgery – after proper and prospective validation of such datasets and applied methods (*Dembek et al., 2019*; *Boutet et al., 2021*; *Elias et al., 2021*). On the hardware side, other possible integrations to the platform in the future include the use of intraoperative imaging such as CT or X-ray acquired for final verification of electrode placement. Data from these acquisitions could potentially be integrated to further enhance visualizations provided by Lead-OR.

The tools, methods, and software described here are not approved by any regulatory authorities and are not intended to assist in making clinical decisions. Rather, we present them for use for purely research-driven purposes under proper IRB approval in study contexts. The tool should be seen as a data visualization tool that could potentially save researchers time by showing data from multiple sources in direct synopsis to one another. As such, it may be powerful to further explore the interplay between electrophysiology and imaging, validate biophysical models, and better characterize patient-specific data.

We see special value in integrating MER-derived measures to the anatomical realm and in the integration with imaging findings. Our aim was to produce a set of use cases each of which could open larger windows of opportunities for upcoming studies. For instance, we included two MER processing pipelines in this study, which have previously been studied in different publications (*Koirala et al., 2020*; *Boëx et al., 2018*; *Zaidel et al., 2009*). The reason for their adoption was mostly demonstrative, and we do not claim for them to be the best/only choices when it comes to studying STN activity. Future work involves analyzing differences in these and similar processing pipelines to derive a better understanding of MER physiology. Given the open-source nature of this project, it will be feasible to extend usability and incorporate complementary approaches. Lead-OR should be seen as a satellite application to existing intraoperative electrophysiology software tools, not an attempt to replace them. The aim of our application is to augment these existing tools by a projection of recorded signals to anatomical space. It is intended to run in parallel to existing software (either on a secondary machine or on the same computer). Hence, thorough inspection and analysis of electrophysiological signals will remain unchanged for users of existing software, while our tool could hopefully add additional insights into the anatomical origins of recorded signals.

In a similar vein, we see larger potential in the field of activation tractography by studying stimulation spread across brain tissue with biophysical models that could range around varying degrees of complexity (*Butenko et al., 2020*; *Gunalan et al., 2017*; *Howell et al., 2019*; *Noecker et al., 2021*). Differences in connectomes (*Horn and Blankenburg, 2016*) vs. pathway atlases (*Petersen et al., 2019*; *Alho et al., 2019*; *Middlebrooks et al., 2020*) vs. individual tractography (*Akram et al., 2017*) acquired in the specific patient could be investigated directly within the operation theater. We foresee that such studies could lead to a better understanding of the mechanism of action of DBS. This study for now showcases this application of visualizing test stimulations in limited and anecdotical form (also see *Figure 4—video 1*), warranting further investigation and validation.

Finally, we see large potential in the use and further aggregation of ultra-high-resolution atlas data. Already, such datasets have been emerging and incorporated into DBS applications (*Edlow et al., 2019*; *Horn et al., 2017*). However, we foresee additional datasets that may revolutionize our definition of anatomy and brain connectivity in the future. For instance, the Jülich group has announced an upcoming version of the BigBrain dataset (*Figure 7*, *Figure 4—video 1*) that will be available in 1 µm resolution (*Horn, 2021*). A recent normative diffusion-MRI connectome available in 760 µm resolution was based on a 9-hr-long scan of a living human brain (*Wang et al., 2021*). Similarly, a structural brain template of the human brain available in 100 µm resolution was acquired by scanning a postmortem brain over 100 hr at $7\,T$ (*Figure 7*; *Edlow et al., 2019*). A recently published pathway atlas of the basal ganglia used expert knowledge and insights from animal studies to create the most realistic set of subcortical fibers available to date (*Petersen et al., 2019*). Similar applications involve histological mesh tractography – a novel technique to create accurate tract representations based on histological data (*Alho et al., 2021*) or expert-curated sets of fiber bundles created by tractography on diffusion MRI data from 1000 subjects (*Middlebrooks et al., 2020*). We foresee great use of such resources if the process of registering them to patient space is truly accurate. The WarpDrive method presented here could embody a missing link in the evolution of making co-registration methodology as precise as possible – with specific focus on regions of particular interest (such as the DBS target zone in our application). For instance, if our aim was to overlay the BigBrain atlas to support our anatomical

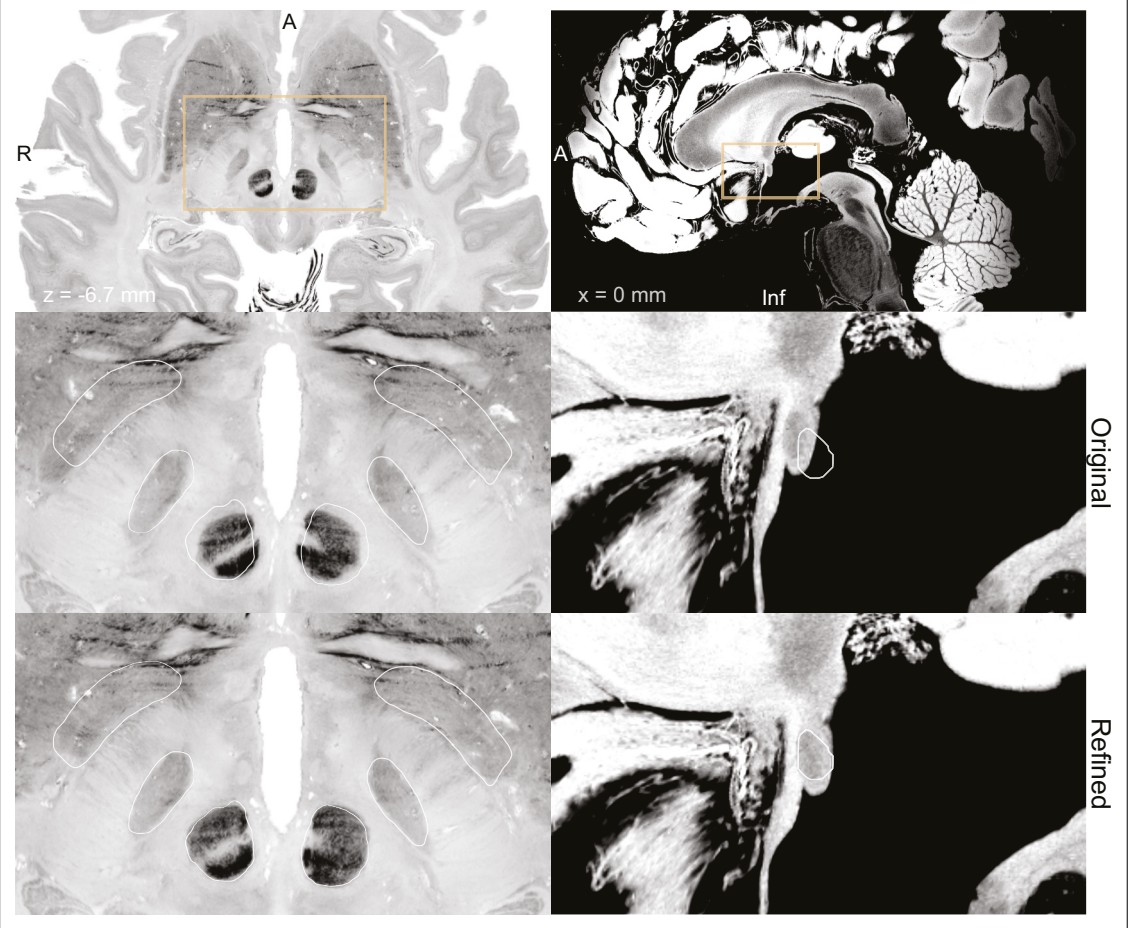

**Figure 7.** BigBrain (*Amunts et al., 2013*; *Xiao et al., 2019*) (left) and 7 T MRI ex vivo human brain template (*Edlow et al., 2019*) (right) are two high-resolution (100 µm isotropic) imaging resources that can be used from Lead-DBS and Lead-OR. The middle panel shows a closeup in a plane using the original transformation to MNI space (Xiao et al. in the case of BigBrain) with white outlines of DISTAL atlas (*Ewert et al., 2018*) (left) and anterior commissure from *Neudorfer et al., 2020* (right). The bottom panel shows the same slice, but using a refined transformation using Lead-DBS and WarpDrive. The refined transformation files can be found in the supplementary data repository (*Oxenford, 2022a*).

knowledge within and around the STN, it is of crucial importance that the registration between atlas and patient imaging of the STN area is meticulously precise. Instead, registration accuracy of, for example, the parietal lobe will be of lesser importance in this particular scenario. WarpDrive gives the user the necessary toolkit to realize highly precise warps, while focusing on specific ROIs (*Figure 7*, *Figure 2—video 1*).

Our results demonstrate general agreement between imaging and electrophysiology data on a group level. The recordings throughout the trajectories present a region with higher activity coinciding with the imaging-based STN. However, as can be seen in *Figure 5*, the agreement is not 100%. Namely, we can observe the presence of activity and high neuronal density in some locations outside of our image-derived model of the STN and vice versa (we observe no activity within voxels that form part of the STN). This emphasizes the possibility of a lack of congruence between preoperative imaging and intraoperative electrophysiological delineation of the STN. Some of these discrepancies could be explained physiologically, for example, seeing activity from regions other than the STN (i.e., thalamic recordings that may be encountered dorsal to the STN or recordings from substantia nigra ventrally). However, true mismatch of the two sources of information (imaging and electrophysiology) *in some cases* is indeed something we would *expect*. Namely, we should not forget that the tool is entirely designed to facilitate integration and visualization of *different* sources of information in parallel. If both would perfectly agree in each single case, there would be no need to acquire MER data in the first place. In our brain shift analysis, we could demonstrate that some of these discrepancies are

associated with the occurrence of brain shift. This presented analysis could be considered a first of its kind attempt to infer brain shift during surgery using a combination of preoperative MRI and intraoperative MER. Specifically, the cross-correlation-derived features may be used as indicators (provided by the program) to quantify discrepancies between MER and imaging data in a real-time setting. This analysis can be further elaborated upon and integrated into future iterations of the platform.

## Limitations

Other explanations for disagreement between imaging and electrophysiological data will directly inform limitations that apply to this study. The occurrence of brain shift could be seen as a limitation but also as a feature of our approach (see above). However, true limitations may arise from imprecisions of the imaging pipeline itself. Although a dedicated multispectral imaging pipeline was applied (in the form of Lead-DBS software), which has shown to create meaningful models of DBS in various studies, there will always be a certain degree of imprecision that is unavoidable when using imaging to segment subcortical nuclei. Here, we aimed to further minimize this imprecision by introducing the WarpDrive tool. However, a downside of this tool could be seen in the fact that it involves manual and observer-dependent steps. Detailed anatomical knowledge and optimal imaging quality are needed to achieve maximal registration accuracy. Ideally, multispectral sets of preoperative images that include specialized sequences optimized for the basal ganglia should be used (*Krauss et al., 2021*). Use of ultra-high-field (i.e., $7\,T$) imaging could represent a useful alternative (*Forstmann et al., 2017*), but in this case danger could arise from increased distortion artifacts exactly and especially in the center of the brain (*Neumann et al., 2015*). Hence, as in the procedure of DBS surgery itself, optimal imaging data quality and meticulous use of tools, as well as optimal levels of methodological insights, are needed to assure safe and successful applications.

Finally, the MER analysis also comes with limitations. First, as the data was collected in retrospective fashion, durations of recordings and distances in recording steps when advancing towards the target were not exactly consistent throughout the whole dataset. Second, cardioballistic artifacts, as well as gradual displacement of brain tissue leading to attenuation of spike amplitudes, are recognized problems when applying spike-sorting algorithms in general. Moreover, anesthesia and wakefulness of patients have an impact on the recordable neurophysiological activity patterns and should be considered when making assumptions about the relationship between neuroanatomy and neurophysiology. While here patients were awake in general, this followed periods of anesthesia (following the clinical protocol established at our center). This leads to a nonuniform quality of the recordings that may then present challenges in their interpretation and processing via automatic algorithms. However, we operate in an experienced high-volume DBS center where surgical decisions are made based on the data used here. In other words, signal quality was sufficient for expert-based decision-making. In the future, additional automatic EEG and EMG activity analysis could further augment the validity of the approach. In general, however, the main aim of this article was to demonstrate the use and feasibility of the tool, while dedicated analyses investigating specific neuroscientific questions should take aforementioned nuances into consideration further.

## Conclusion

We presented a method and open-source software tool to visualize results derived from MERs in anatomical space, together with information derived from patient-specific MRI data, as well as high-resolution atlas resources during DBS surgery. We demonstrated general agreement between imaging and electrophysiology-derived measures, as well as examples of unavoidable discrepancy between the two modalities. The tool has potential to empower scientific studies investigating several topics outlined in our discussion, as well as high potential for clinical translation and represents a first step to help integrate information across sources within two- and three-dimensional visualization scenes. While the software is not certified and intended for scientific use under IRB approval only, subsequent steps will involve improving and extending the different components of the software to achieve a reliable multimodal patient-specific navigator capable of assisting clinical decision-making.

## Data availability

All processed data and code needed to reproduce the main findings of the study are made openly available in deidentified form (see figure legends). This can be found in https://github.com/simonoxen/

Lead-OR_Supplementary, (copy archived at swh:1:rev:c7b8661f0587db992e7eba978d61da8c-d7cdc88b; *Oxenford, 2022a*) and attached to the publication. Due to data privacy regulations of patient data, raw data cannot be publicly shared. Upon reasonable request to the corresponding author, data can be made available after setting up a data-sharing agreement between our host institution (Charité – Universitätsmedizin Berlin) and the inquiring party. All codes used to analyze the dataset are available within Lead-DBS/-OR software (https://github.com/netstim/leaddbs [*Network Stimulation Laboratory, 2022*]; https://github.com/netstim/SlicerNetstim [*Oxenford, 2022b*]).

## Acknowledgements

We thank Alaa Hanna from Alpha Omega Engineering for methodological support in interfacing with the NeuroOmega SDK. While the present software implementation works with two commercial products for surgical planning and the intraoperative procedure, the choices of these systems were arbitrary (defined by what was present at our center) and do not mean any form of endorsement whatsoever. No industry funding was received to carry out this study. Part of this study was presented and worked on during the 35th NA-MIC Project Week (Kapur et al., 2016). We would like to thank the NA-MIC community, especially Dr. Andras Lasso and Dr. Steve Pieper for the help and discussions on 3D Slicer modules implementation. AH was supported by the German Research Foundation (Deutsche Forschungsgemeinschaft, Emmy Noether Stipend 410169619 and 424778381 – TRR 295), Deutsches Zentrum für Luft- und Raumfahrt (DynaSti grant within the EU Joint Programme Neurodegenerative Disease Research, JPND), the National Institutes of Health (2R01 MH113929), as well as the Foundation for OCD Research (FFOR). AH is participant in the BIH-Charité Clinician Scientist Program funded by the Charité – Universitätsmedizin Berlin and the Berlin Institute of Health. WJN was supported by Bundesministerium für Bildung und Forschung (BMBF) (Project iDBS FKZ01GQ1802), Deutsche Forschungsgemeinschaft (DFG) (Project ID 424778371 – TRR 295), Hertie Foundation (Project BGPlasticity), and Berlin Institute of Health (Project SPOKES).

## Additional information

### Competing interests

Andrea Kühn: Reports personal fees from Medtronic, Boston Scientific, Abbott, Teva, Ipsen and Stadapharm, all outside the submitted work. Andreas Horn: Reports lecture fee for Boston Scientific outside the submitted work. The other authors declare that no competing interests exist.

### Funding

| Funder | Grant reference number | Author |
|---|---|---|
| Deutsche Forschungsgemeinschaft | Emmy Noether Stipend 410169619 | Andreas Horn |
| Deutsche Forschungsgemeinschaft | Project ID 424778371 | Andreas Horn Wolf-Julian Neumann |
| Bundesministerium für Bildung und Forschung | Project iDBS FKZ01GQ1802 | Wolf-Julian Neumann |
| Deutsches Zentrum für Luft- und Raumfahrt | DynaSti grant within the EU Joint Programme Neurodegenerative Disease Research JPND | Andreas Horn |
| National Institutes of Health | 2R01 MH113929 | Andreas Horn |

The funders had no role in study design, data collection and interpretation, or the decision to submit the work for publication.

### Author contributions

Simón Oxenford, Conceptualization, Data curation, Formal analysis, Investigation, Methodology, Software, Visualization, Writing – original draft, Writing – review and editing; Jan Roediger,

Conceptualization, Data curation, Formal analysis, Methodology, Writing – original draft, Writing – review and editing; Clemens Neudorfer, Conceptualization, Investigation, Methodology, Writing – review and editing; Luka Milosevic, Conceptualization, Formal analysis, Methodology, Writing – review and editing; Christopher Güttler, Methodology, Writing – review and editing; Philipp Spindler, Data curation, Methodology, Writing – review and editing; Peter Vajkoczy, Wolf-Julian Neumann, Andrea Kühn, Supervision, Writing – review and editing; Andreas Horn, Conceptualization, Formal analysis, Funding acquisition, Investigation, Methodology, Project administration, Software, Supervision, Visualization, Writing – original draft, Writing – review and editing

### Author ORCIDs
Simón Oxenford  http://orcid.org/0000-0003-2989-3861
Jan Roediger  http://orcid.org/0000-0003-2814-3532
Luka Milosevic  http://orcid.org/0000-0002-4051-5397
Wolf-Julian Neumann  http://orcid.org/0000-0002-6758-9708
Andreas Horn  http://orcid.org/0000-0002-0695-6025

### Ethics
The collection and analysis of all patient data used for this article was approved by the Local Ethics committee of Charité - Universitätsmedizin Berlin (master vote EA2/145/21). All data were analyzed retrospectively and obtained in deidentified from Medical Records of Charité. Hence, following local guidelines in Berlin/Brandenburg as well as NIH guidelines for human subjects research, no explicit patient consent to analyze and publish was obtained/necessary.

### Decision letter and Author response
Decision letter https://doi.org/10.7554/eLife.72929.sa1
Author response https://doi.org/10.7554/eLife.72929.sa2

---

## Additional files

### Supplementary files
• Transparent reporting form
• Supplementary file 1. Inclusion flow chart.
• Reporting standard 1. STROBE checklist.

### Data availability
All processed data and code needed to reproducemain findings of the study aremade openly available in de-identified form (see figure legends). This can be found in https://github.com/simonoxen/Lead-OR_Supplementary (copy archived at swh:1:rev:c7b8661f0587db992e7eba978d61da8cd-7cdc88b). Due to data privacy regulations of patient data, raw data cannot be publicly shared. Upon reasonable request to the corresponding author, data can be made available after setting up a data sharing agreement between our host institution (Charité - Universitätsmedizin Berlin) and the inquiring party. All code used to analyze the dataset is available within Lead-DBS /-OR software (https://github.com/netstim/leaddbs; https://github.com/netstim/SlicerNetstim(copy archived at swh:1:rev:2439c1e117af9027802ba48b67530a0af189c6fe)).

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
