## [Editor Report]

The authors present a software tool combining and correlating the documentation of intraoperative neurophysiological findings with atlas and imaging data. They also show an exemplary validation of their tool in a clinical series of 52 Parkinson's disease patients who underwent DBS surgery. This article will be of interest to clinicians and researchers who are involved in both the placement and controlling of the accuracy of the location of deep brain stimulation electrodes.

---

## [Decision Letter]

**Decision letter after peer review:**

Thank you for submitting your article "Lead-OR: A Multimodal Platform for Deep Brain Stimulation Surgery" for consideration by *eLife*. Your article has been reviewed by 2 peer reviewers, and the evaluation has been overseen by a Reviewing Editor Lars Timmermann and Christian Büchel as the Senior Editor. The following individual involved in review of your submission has agreed to reveal their identity: Ausaf Bari (Reviewer #2).

This paper from Oxenford et al. describes a new open-source software tool LEAD-OR which allows a real-time visualization and merging of imaging and MER data during a DBS implantation procedure.

The guest editor and the reviewers were agreeing that this new technology is a major step in academic analysis of joint electrophysiological and imaging derived information for the analysis of target areas for deep brain stimulation. However, the reviewers also see some points that should be made a little more clear to envision the potential pitfalls and limitations of this program:

1. MER data has been highly selectively analyzed retrospectively. A number of patients were sorted out because of insufficient MER data. Could the authors please give an impression how the program gives feedback to investigators in a real-life OR situation about e.g. noisy MER recordings? Does the program give a sufficient feedback? How are processing times during the OR approach using rea-time MER?

2. How is the actual visualization while the program is running in the OR?

3. The authors mention the brain shift as the major source of discrepancy between imaging data and MER data. However, those are the patients who really need a good localization by MER and a "warning" to the OR team that imaging based on pre-OP MRI is not reflecting the actual situation anymore…. With other words: We would like to see from the authors how the program visualizes in the patients with a larger brain shift the critical discrepancies between MER localization and imaging.

4. In many ORs intraoperative imaging is used to verify the correct electrode position. In some this is done by the AERO-CT, in some ORs a fixed conventional stereotactic imaging set-up is used. How is this intra-operative imaging integrated into LEAD-OR?

5. It would be nice to see a number of single cases with unusual STN configurations. The value of this program will NOT be the correct classification and identification in the classical textbook cases but rather in the patients with unusual configurations of e.g. the STN.

6. We are not totally clear in understanding: What is really the GOLD-Standard in defining the target area in the visualization for the OR team? The MER? OR pre-OP imaging? Or intra-OP imaging?

Please also have a detailed look at the reviewers comments and revise carefully.

*Reviewer #1 (Recommendations for the authors):*

There are several issues which need to be addressed upon revision of the manuscript:

The statement in the abstract that one would use "commonly … up to five trajectories in parallel" is inconclusive as it stands. Better to say, "using a single trajectory or up to five trajectories in parallel".

Page 2, line 41. The frame itself does not include markers. Such markers are located in the fiducial plates which are mounted for imaging.

Page 2, line 55: Most "expert electrophysiologists" actually are "expert neurosurgeons or neurologists".

Page 3, line 100. Why should the use of Lead-OR be "strictly limited to Institutional Review Board (IRB) approved Research"? This does not really make sense since IRB approval would not address questions of liability associated with open-source software tools. The discussion with that regard is also misleading. "Study contexts" do not solve the inherent problems.

It is good to see that the tool can be applied when several microelectrode recording trajectories are used in parallel, but it would be better to discuss its feasibility first for a single trajectory and then for three or up to five.

Figure 1 is quite unclear. The red nucleus appears to be 10 times larger than the STN in this figure, also the colours appear to be wrongly applied. Please correct.

In general, I would recommend not to use too many abbreviations in figure legends. What does SDK mean?

Figure 6 is referred to prior referring to figures 4 and 5. Please correct.

It is a weakness of the patient series that some recordings were made under general anaesthesia. The problems with this should be outlined more clearly.

What type of "macroelectrodes" were used?

*Reviewer #2 (Recommendations for the authors):*

This is an exciting achievement by the authors. However, my main criticism is that it is a fairly descriptive paper without a central scientific premise. It would have been strengthened if for example the STN recordings and atlas overlays had been analyzed in more depth as that could have served as the scientific premise in this case. The data would also be further bolstered by including thalamic and GP cases.

[Editors' note: further revisions were suggested prior to acceptance, as described below.]

Thank you for resubmitting your work entitled "Lead-OR: A Multimodal Platform for Deep Brain Stimulation Surgery" for further consideration by *eLife*. Your revised article has been evaluated by Christian Büchel (Senior Editor), a Reviewing Editor, and three reviewers.

The manuscript has been improved but there are some remaining issues that need to be addressed, as outlined below:

*Reviewer #3 (Recommendations for the authors):*

Summary:

Oxenford and colleagues present a novel framework for integrating data from multiple domains during surgery for Deep Brain Stimulation lead placement to enable informed decisions about the best target location in key regions associated with Parkinson's disease in the future. The innovative approach of the very diligent group from Berlin goes far beyond the common practice of implantation based on the assessment of imaging results and neurophysiological patterns encountered and the team's experience in parallel and towards an integrative and publicly available toolbox aimed at processing this information at once.

Review:

The approach from the group paves the way for the discussion of how to better objectify target selection away from the sole consideration of often imperfect imaging by extending it via intraoperative recordings – what some would consider the undisputed gold standard for DBS lead placement. It's a very well written and meticulously ornated manuscript with multitudes of details and visualisations. One of the intriguing and possibly most underestimated parts of this work is its open source character, which lends credence to the analyses but invites literate scientists to obtain deep insights into technical details; but at the same time also offers the possibility to include this toolbox easily into their routine. This open science approach is a lived practice for this group as an unparalleled service for the entire DBS community. In the referee's opinion, it links at the same time a potential drawback for other centres with less experience. Regardless of the multitude of potential extensions in principle (tractography, intraoperative imaging, etc.), it remains to be ascertained whether centres with less experience and technical know-how will be able to incorporate such a toolbox into their protocols. Nevertheless, an overly specialized solution with a high potential for expansions should not serve as excessive criticism of the authors' highly innovative concept and the well-balanced manuscript.

Recommendations for the authors:

From a personal point of view, the reviewer would preferably arrange Figure 5E the other way round, as this is more intuitive to understand. In any case, it should be briefly noted that according to RAS, with increasing values on the z-axis, the layers display more dorsal/superior slices. This figure is yet also quite interesting in terms of its content, as it implies that there is a great overlap between imaging and putative neurophysiologic epiphenomena of the STN in PD-patients but at the same time offers a somewhat puzzling result of a discrepancy towards (a) the location of the dorsolateral STN and (b) to what is generally believed to be the spot to aim for. The reviewer does believe Oxenfurt et al. have already done a marvellous job adding information and addressing points that have been arisen, nonetheless, inclusion of considerations into the discussion of where final electrodes were located, ergo where the best clinical outcomes were to be found and the proposed combined analyses seem interesting the least.

One of the aspects that needs further explanation is the number of artefacts or that of participants. On the one hand, it is undisputed that the exclusion of erroneous or low-quality data is a common practice. Nonetheless, 40% exclusions appear rather high, so this may need some further clarification, as possibly indicated by another reviewer. In addition, the number of 52 patients included does not seem very transparent. It would be helpful to explain where these people came from, i.e. whether they were consecutive patients within a period of time or specially selected people, etc.

---

## [Author Response]

The reviewers have discussed their reviews with one another, and the Reviewing Editor has drafted this to help you prepare a revised submission.1. MER data has been highly selectively analyzed retrospectively. A number of patients were sorted out because of insufficient MER data. Could the authors please give an impression how the program gives feedback to investigators in a real-life OR situation about e.g. noisy MER recordings? Does the program give a sufficient feedback? How are processing times during the OR approach using rea-time MER?

As the editor mentioned, we manually discarded data with insufficient electrophysiological or imaging quality to perform our analysis. However, especially in the electrophysiological domain, we believe it to be quite common (e.g., post-processing of most EEG experiments include a step of epoching where episodes with artifacts are discarded). We are unaware of an algorithm that could reliably replace expert-based epoching to do so.

In the real-time setting, it is crucial to note that Lead-OR is a satellite program to the actual electrophysiology software used (e.g. the NeuroOmega workstation). It is not the intention of Lead-OR to replace these systems, but rather to complement them with information from neuroimaging. Users can still see raw data from acquisition devices and based on their knowledge and experience make an understanding of the quality of the signals. Hence, in this first iteration of the program we did not include robust routines to quantify signal quality. Not even the official electrophysiological software tools (like NeuroOmega or Leadpoint) do so.

This being said, we now made it is possible to remove data points calculated by the program in a real-time setting. For example, if users would identify poor quality recordings which then derived non-sensical measurements by the program, they are now able to discard them in order not to compromise the visualization.

The computation of features takes a few seconds and is continuously running to update the visualization. Again, this very much compares to other algorithms, such as the HaGuide tool issued by AlphaOmega. We have now further highlighted this in the real-time setting video (Figure 4-video 2; see attached to the submission) introduced in the next point. As can be seen in the following points, we now include a large number of additional cases, unusual cases, videos, and examples with GPi/ thalamic surgery to illustrate how Lead-OR performs in different clinical scenarios. We added the following paragraphs to further clarify aforementioned points:

“The computation of features is continuously executed for each position of the microdrive, updating the recording stream at each time point. This process takes a few seconds (depending on available hardware), and the visualization is then updated.” – methods, p. 6

“Lead-OR should be seen as a satellite application to existing intraoperative electrophysiology software tools, not an attempt to replace them. The aim of our application is to augment these existing tools by a projection of recorded signals to anatomical space. It is intended to run in parallel to existing software (either on a secondary machine or on the same computer). Hence, thorough inspection and analysis of electrophysiological signals will remain unchanged for users of existing software, while our tool could hopefully add additional insights about the anatomical origins of recorded signals.” – discussion, p. 13

2. How is the actual visualization while the program is running in the OR?

To demonstrate the visualization of the surgical procedure we have included an additional video (Figure 4-video 2; see attached to the submission) in the revised version of the manuscript. Specifically, we highlight Lead-OR’s user interface and provide a thorough walkthrough demo of how Lead-OR is set up and executed intraoperatively including connection to the hardware device, channel selection for recording, configuration of trajectories, input planning, stimulation set up, visualization and processing of live data streaming into the software. The added legend reads:

“Figure 4-video 2. Video showing the program user interface and its use. Includes exemplary connection to the hardware device, channel selection for recording, configuration of trajectories, input planning, stimulation set up, visualization and processing of live data streaming into the software.” – results, p. 10

3. The authors mention the brain shift as the major source of discrepancy between imaging data and MER data. However, those are the patients who really need a good localization by MER and a "warning" to the OR team that imaging based on pre-OP MRI is not reflecting the actual situation anymore…. With other words: We would like to see from the authors how the program visualizes in the patients with a larger brain shift the critical discrepancies between MER localization and imaging.

We absolutely agree. However, we would also like to stress the following point: As the editor mentions, if anatomical and electrophysiological information were to always converge, we would need only one of the two. Hence, disagreement between the two information sources — and visualization of this disagreement — may be leveraged to provide information regarding size and extent of anatomical variability. Without overlaying the two sources of information in anatomical space (standard care), this mismatch goes unnoticed.

We were thrilled by the idea of the editor to add a warning and deliberate information about brain shift occurring during surgery. This took considerable effort since a novel algorithm was needed, but we believe we can show something promising. We calculated the displacement in each trajectory at which the imaging and electrophysiology-derived measures most corresponded to each other. In our sample, these displacements (cross-correlation lags) correlated with brain shift as observed on *postoperative* CT.

The following sections and a novel figure were amended in the revised version of the manuscript:

“Using the Lead-DBS pipeline we carried out brain shift correction using post-operative computerized tomography data (Horn et al., 2019). This allowed us to quantify the degree of brain shift occurring after surgery based on imaging derived metrics. For each trajectory, each recording position was displaced using the brain shift correction transform. We took the median of displacements as the amount of brain shift for each trajectory. We will refer to this measure as the imaging-based brain shift estimate in this manuscript (note that it is derived from pre- and post- operative imaging data).” – methods, p. 8

“Additionally, NRMS values and STN distances for each trajectory were transformed with the inverse tangent function resulting in similar distributions of the two. Subsequent cross correlation of these two signals along each trajectory resulted in a maximum cross correlation value and the lag (displacement) at which this maximum occurred (Figure 5–source code 2).” – methods, p. 9

“In a next step, we sorted the trajectories according to their maximum cross correlation and split the data in half, retaining the trajectories that were in close proximity to the STN and showed electrophysiological activity. We then sorted the top half according to the lag at which the maximum cross correlation occurred (Figure 5–source code 2). We will refer to this lag as the electrophysiology-based brain shift estimate (note that it is derived from pre-operative imaging and intra-operative electrophysiology). Hence, in contrast to the imaging-derived brain shift estimate (which required postoperative imaging), this one could be computed during surgery. The electrophysiology-based brain shift measures were compared to the imaging-based brain shift estimates in two ways: First, we contrasted imaging-based brain shift estimates between the low versus high lag groups using Wilcoxon’s signed rank test. The high lag group was defined by taking trajectories with lag values above one standard deviation of the lags. The low lag group is composed by the same number of trajectories taken from the data sorted according to the lag. This would analyze whether cases with high electrophysiology-derived estimates indeed had more brain shift based on the imaging-based estimate. Second, we correlated values from the high lag trajectories (where significant brain shift was estimated based on electrophysiology) with the imaging-derived estimate of brain shift. This would analyze whether the degree of brain shift would correlate between electrophysiology- and imaging-derived estimates.” – methods, p. 9

“With respect to the brain shift analysis, the low lag and high lag groups showed a significantly different brain shift distribution (Wilcoxon’s signed rank test ρ = 0.0076). Also, correlating the high lag values (electrophysiology-derived brain shift estimate) with their imaging-derived brain shift estimates showed a significant association (R = 0.40, ρ = 0.016; Figure 5—figure supplement 1). Figure 5—figure supplement 2 shows an example case illustrating how the imaging-based brain shift corrected Lead-OR scene presents better correspondence between imaging and MER.” – results, p. 10

“In our brain shift analysis we could demonstrate that some of [the imaging and electrophysiology] discrepancies are associated with the occurrence of brain shift. This presented analysis could be considered a first of its kind attempt to infer brain shift during surgery using a combination of preoperative MRI and intraoperative MER. Specifically, the cross correlation derived features may be used as indicators (provided by the program) to quantify discrepancies between MER and imaging data, in a real-time setting. This analysis can be further elaborated upon and integrated in future iterations of the platform.” – discussion, p. 15

4. In many ORs intraoperative imaging is used to verify the correct electrode position. In some this is done by the AERO-CT, in some ORs a fixed conventional stereotactic imaging set-up is used. How is this intra-operative imaging integrated into LEAD-OR?

While it would be interesting and beneficial, currently, such a feature is not integrated into Lead-OR. We thought long and hard about whether we could easily integrate something during the revision, but concluded it would be out of scope for the current manuscript. The main reasons are as follows: First, as the editor mentions, different sources from 2D and 3D imaging are used and especially registering 2D imaging to a 3D scene is an unsolved research field of its own. Second, not even commercial real-time applications (such as Brainlab Elements) do not include such a feature that is intended for intraoperative use. Third, while related to the aim of Lead-OR, the use of intraoperative imaging should be seen as tangential. While the aim of Lead-OR was to fuse electrophysiology with intraoperative imaging, the reason to acquire intraoperative imaging is to confirm electrode placement within the intended target. However, while we had to take a rain check at present, we believe that the inclusion of Lead-OR as an open-source tool within the 3D Slicer environment has the potential to further include the features the editor had in mind. Indeed, multiple other plugins of 3D Slicer could be used/recruited and combined with Lead-OR to flexibly adapt to the needs in each OR.

We added the following to the discussion clarifying this point:

“On the hardware side, other possible integrations to the platform in the future include the usage of intra-operative imaging such as CT or X-ray acquired for final verification of electrode placement. Data from these acquisitions could potentially be integrated to further enhance visualizations provided by Lead-OR.” – discussion, p. 13

5. It would be nice to see a number of single cases with unusual STN configurations. The value of this program will NOT be the correct classification and identification in the classical textbook cases but rather in the patients with unusual configurations of e.g. the STN.

We agree that inclusion of cases with unusual configurations will be useful for the reader to better grasp the capabilities of Lead-OR. Thus, we have amended the manuscript and now provide additional figures highlighting how Lead-OR handles unusual anatomical configurations (i.e., narrow skull, asymmetric hemispheres, broad skull) as well as different targets (i.e., VIM and GPi) as suggested by reviewer #2. The *Results section* of the manuscript was amended as follows:

“Furthermore, Figure 4—figure supplement 1 shows the application of the tool in a VIM and GPi example. Finally, for illustrative purposes, we included additional three STN cases with unusual anatomical configurations in Figure 4—figure supplement 2.” – results, p. 10

6. We are not totally clear in understanding: What is really the GOLD-Standard in defining the target area in the visualization for the OR team? The MER? OR pre-OP imaging? Or intra-OP imaging?

We would argue that there is no clear gold-standard and for this reason multiple sources of information are usually combined to form an expert decision (imaging, MER, LFP, test stimulations). Furthermore, the strategy of the surgical team may differ from institution to institution. As the editor surely knows, some centers solely rely on pre-operative imaging (as e.g., the London group or the team at Brigham & Women’s), others carry out MER recordings and test stimulations, etc.

In our opinion, the introduced tool could be a crucial help to exactly form this decision that relies on multi-modal data by fusing key metrics of all data sources in a single three-dimensional scene. However, we must emphasize that the target area is defined by the surgeon in stereotactic planning software (e.g., Brainlab in our case). Lead-OR does not propose a target implantation area. Instead, it provides means to visualize data to facilitate decisions by the surgical team.

We the following paragraph was modified to make this clearer:

*“*The tools, methods and software described here are not approved by any regulatory authorities and are not intended to assist in making clinical decisions. Rather, we present them for use in purely research driven purposes under proper IRB approval in study contexts. The tool should be seen as a data visualization tool that could potentially save researchers time by showing data from multiple sources in direct synopsis to one another. As such, it may be powerful to further explore the interplay between electrophysiology and imaging, to validate biophysical models and to better characterize patient specific data.*” –* discussion, p. 13

Please also have a detailed look at the reviewers comments and revise carefully.Reviewer #1 (Recommendations for the authors):There are several issues which need to be addressed upon revision of the manuscript:The statement in the abstract that one would use "commonly … up to five trajectories in parallel" is inconclusive as it stands. Better to say, "using a single trajectory or up to five trajectories in parallel".

Thank you for raising this point. Unrelated to this comment, we revised the abstract to make it clearer and removed the mention to the number of trajectories. The *background* section in the abstract now reads:

*“*Deep Brain Stimulation (DBS) electrode implant trajectories are stereotactically defined using preoperative neuroimaging. To validate the correct trajectory, microelectrode recordings (MER) or local field potential recordings (LFP) can be used to extend neuroanatomical information (defined by magnetic resonance imaging) with neurophysiological activity patterns recorded from micro- and macroelectrodes probing the surgical target site. Currently, these two sources of information (imaging vs. electrophysiology) are analyzed separately, while means to fuse both data streams have not been introduced.*” –* Abstract, p. 1

Page 2, line 41. The frame itself does not include markers. Such markers are located in the fiducial plates which are mounted for imaging.

Thank you for raising our awareness of this issue. We have revised the *Introduction* as follows:

*“*Surgical planning is usually carried out after fusing the MRI sequences with a computed tomography (CT) volume acquired with the stereotactic frame and fiducial plates already mounted to the patient’s head. The fiducial plates include markers that are used to convert stereotactic coordinates (established in the planning software) to frame coordinates (applicable to mechanically adjust the stereotactic frame) in order to place electrodes to the intended target.*”* – introduction, p. 2

Page 2, line 55: Most "expert electrophysiologists" actually are "expert neurosurgeons or neurologists".

We thank the reviewer for this comment. We have amended the manuscript as follows:

*“*While most centers analyze MERs by visual and auditory inspection from expert neurosurgeons or neurologists, the first FDA and CE-approved machine-learning algorithms that facilitate this monitoring step have recently been introduced, *…” –* Introduction, p 2

Page 3, line 100. Why should the use of Lead-OR be "strictly limited to Institutional Review Board (IRB) approved Research"? This does not really make sense since IRB approval would not address questions of liability associated with open-source software tools. The discussion with that regard is also misleading. "Study contexts" do not solve the inherent problems.

The reviewer raises an important point. While the legal framework within and around the use of research software for medical research may differ between countries, we want to emphasize that Lead-OR is neither CE marked nor FDA approved software. Similar tools, such as the StimVision (Noecker et al. 2021) are non-approved software tools which clarify that their use is “strictly limited to Institutional Review Board (IRB) approved research studies at individual academic institutions”. This being said, we defer to local laws and review board practices on whether the use of Lead-OR within a study context can or cannot be approved. The manuscript was amended as follows:

*“*Lead-OR is intended for purely academic research use and does not have any form of government body regulatory approval. As such, any use of Lead-OR is strictly limited to Institutional Review Board (IRB) approved research studies at individual academic institutions, while legal frameworks and practices may differ from country to country.*” –* methods, p. 3

We would argue that study contexts do indeed “solve” the problem in that the regulatory authority would be an IRB (or similar depending on legal context), and not the FDA / CE. For instance, the aforementioned StimVision tool was used in multiple studies that involved surgery for depression (e.g., Riva-Posse 2017 Mol Psych).

In most legal contexts that we are aware of, IRB approval would be needed e.g. to prospectively study the use and safety of such a novel software. In Germany, however, IRBs would need further approval by the Landesgesundheitsamt to give their consent (device study).

In other words, with this passage we wanted to state as clearly as possible to all readers that we are not introducing a certified software but a research software, and it will be the task of interested users to make sure they are allowed by local regulatory bodies to carry out what they intend to do.

It is good to see that the tool can be applied when several microelectrode recording trajectories are used in parallel, but it would be better to discuss its feasibility first for a single trajectory and then for three or up to five.

We would like to respectfully disagree with this point and cannot completely see the advantage of analyzing data on one-trajectory procedures, currently. Maybe the reviewer could clarify how or why this would lead to advantages, further?

Also, this would change our complete study, since all analyses build on the retrospective (and multi-electrode) data, we currently have. At our center, we do not carry out surgeries with single trajectories. Our local surgeons strongly advised against the use of single trajectories, in the past. Here at Charité, a minimum of 2 trajectories are always applied to maximize stability of the path they enter the brain in x/y directions.

The following was added to clarify this:

“The setup interacts with commercial tools for surgical planning and MERs and has the capability to visualize and analyze data in various forms. In the presented group study, the data acquisition conditions were not controlled for, given their retrospective nature. However, the platform can generalize to alternate settings. For example, the number of trajectories used can be set from one to five, without compromising its execution. With respect to hardware settings, while currently, a fixed set of interfaces to commercial tools is available, the open-source nature of the software will allow integration of links to other devices.” – discussion, p. 12.

Figure 1 is quite unclear. The red nucleus appears to be 10 times larger than the STN in this figure, also the colours appear to be wrongly applied. Please correct.

Thank you for raising this point. We revised the figure to make its appearance clearer.

In general, I would recommend not to use too many abbreviations in figure legends. What does SDK mean?

‘SDK’ stands for ‘Software Development Kit’. We agree that abbreviations should be avoided as much as possible. We revised all figure legends taking this into account.

Figure 6 is referred to prior referring to figures 4 and 5. Please correct.

We thank the reviewer for their attentive comment. We have corrected this mistake.

It is a weakness of the patient series that some recordings were made under general anaesthesia. The problems with this should be outlined more clearly.

We agree with the reviewer that this is a crucial point and a strong limitation of our study. We added the following to our limitations section to further address this:

*“*Moreover, anesthesia and wakefulness of patients have an impact on the recordable neurophysiological activity patterns and should be considered when making assumptions about the relationship between neuroanatomy and neurophysiology. While here, patients were awake in general, this followed periods of anesthesia (following the clinical protocol established at our center). This leads to a non-uniform quality of the recordings which may then present challenges in their interpretation and in their processing via automatic algorithms. However, we operate in an experienced high-volume DBS center where surgical decisions are made based on the data used here. In other words, signal quality was sufficient for expert-based decision making. In the future, additional automatic EEG and EMG activity analysis could further augment the validity of the approach. In general, however, the main aim of the present manuscript was to demonstrate the use and feasibility of the tool, while dedicated analyses investigating specific neuroscientific questions should take aforementioned nuances into consideration further.*” –* discussion p. 15

What type of "macroelectrodes" were used?

We have amended the *Methods* section to clarify which macroelectrodes were used during surgery:

“Neuroprobe Sonus non-shielded microelectrodes (AlphaOmega, Nazareth, ISR) were used as micro/macroelectrodes.” – methods, p. 8

Reviewer #2 (Recommendations for the authors):This is an exciting achievement by the authors. However, my main criticism is that it is a fairly descriptive paper without a central scientific premise. It would have been strengthened if for example the STN recordings and atlas overlays had been analyzed in more depth as that could have served as the scientific premise in this case. The data would also be further bolstered by including thalamic and GP cases.

We thank the reviewer for raising this point. We have included thorough additional analyses on brain shift, added multiple additional case examples (also see comments above) and — based on the reviewer’s suggestion — included an additional figure highlighting exemplary pallidal and thalamic cases. We have amended the *Results* and *Discussion section*s as follows:

“Furthermore, Figure 4—figure supplement 1 shows the application of the tool in a VIM and GPi example. Finally, for illustrative purposes, we included additional three STN cases with unusual anatomical configurations in Figure 4—figure supplement 2.” – results, p. 10

“Furthermore, although we present the tool and analysis made with STN cases, it could also be applied for other DBS targets. As illustrative examples, we refer to Figure 4—figure supplement 1 to see Lead-OR visualizations for a VIM and a GPi case.” – discussion, p. 12

It is important to note that the paper was submitted to the tools and resources section of *eLife*, in which “*articles do not have to report major new biological insights or mechanisms, but it must be clear that they will enable such advances to take place, for example, through exploratory or proof-of-concept experiments”* (https://reviewer.elifesciences.org/author-guide/types). Still, we aimed to answer one hypothesis driven experiment, which addresses the agreement between electrophysiology and imaging-based reconstructions of the STN. In the revised version of the manuscript, we additionally investigate how brain shift could be estimated in real-time using the combination between preoperative imaging and electrophysiological data (refer to point 3 raised above). We apologize if these analyses did not come out as clear in the manuscript. The following modified sections together with the analyses in point 3 were added to clarify this comment.

“The main hypothesis from the group analysis was that electrophysiological recordings acquired from within the imaging defined STN would show higher activity than the ones recorded outside of the STN.” – methods, p. 9

“Comparing the NRMS from the bottom 20% (outside of the STN) to the top 20% revealed an anatomical region with significant differences (ρ < 0.01) within the imaging defined STN boundaries (defined as the median of the top 20% STN boundaries). In other words, the recorded activity from inside this part of the STN were significantly higher than the ones recorded outside of it.” – results, p. 10

“Our results demonstrate general agreement between imaging and electrophysiology data on a group level. The recordings throughout the trajectories present a region with higher activity coinciding with the imaging-based STN.” – discussion, p. 15

[Editors' note: further revisions were suggested prior to acceptance, as described below.]

The manuscript has been improved but there are some remaining issues that need to be addressed, as outlined below:Reviewer #3 (Recommendations for the authors):From a personal point of view, the reviewer would preferably arrange Figure 5E the other way round, as this is more intuitive to understand. In any case, it should be briefly noted that according to RAS, with increasing values on the z-axis, the layers display more dorsal/superior slices. This figure is yet also quite interesting in terms of its content, as it implies that there is a great overlap between imaging and putative neurophysiologic epiphenomena of the STN in PD-patients but at the same time offers a somewhat puzzling result of a discrepancy towards (a) the location of the dorsolateral STN and (b) to what is generally believed to be the spot to aim for. The reviewer does believe Oxenfurt et al. have already done a marvellous job adding information and addressing points that have been arisen, nonetheless, inclusion of considerations into the discussion of where final electrodes were located, ergo where the best clinical outcomes were to be found and the proposed combined analyses seem interesting the least.

We thank the reviewer for their attentive comments and interest in the submitted work. We rearranged panel E in figure 5, now showing decreasing z values from left to right.

Following up with the reviewer’s suggestion, we are excited to include an additional supplementary figure where clinical DBS settings are visualized together with group results of the study shown in figure 5 of the manuscript. Given this was not the main focus of the study, we only introduce a qualitative analysis. A quantitative report describing clinical stimulation with micro electrode recordings features is not feasible with the present retrospective dataset (where clinical outcomes were not taken systematically at a fixed postoperative outcome, across this cohort). We amended the following sections and included an additional supplementary figure.

“The most recent available clinical stimulation settings were retrieved from all 32 patients (visits ranging from 3 to 44 months after surgery). We reconstructed DBS electrodes based on the standard Lead-DBS pipeline and denoted the coordinate of the active contact (in case of multiple active contacts, their locations were averaged). For a qualitative analysis, we projected this coordinate to the nearest point along the closest trajectory for each electrode.” – methods, p 8

“In Figure 5-Suplementary figure 3 we show clinical active contact coordinates with respect to the results of the group analysis as shown in figure 5. Most of the coordinates rely inside the STN and coincide with high activity regions as depicted by the micro electrode recordings.” – results, p 10

One of the aspects that needs further explanation is the number of artefacts or that of participants. On the one hand, it is undisputed that the exclusion of erroneous or low-quality data is a common practice. Nonetheless, 40% exclusions appear rather high, so this may need some further clarification, as possibly indicated by another reviewer. In addition, the number of 52 patients included does not seem very transparent. It would be helpful to explain where these people came from, i.e. whether they were consecutive patients within a period of time or specially selected people, etc.

We thank the reviewer for pointing this out. We explained further how inclusion was carried out in the text and in a more detailed way in a supplementary diagram. We modified the manuscript as follows:

“52 patients (12 female; mean age = 61 ± 9) were retrieved from cases undergoing STN-DBS surgery at Charité — Universitätsmedizin Berlin between 07/2017 until 10/2021. Inclusion was based on having homogeneous data acquisitions consistent with current surgical procedure (i.e., Brainlab planning exports together with corresponding imaging acquisitions and complete micro electrode recording information). Supplementary file 1 summarizes the inclusion process in form of a flow-chart.” – methods, p. 8